# Loops around the Heme Pocket Have a Critical Role in the Function and Stability of *Bs*DyP from *Bacillus subtilis*

**DOI:** 10.3390/ijms221910862

**Published:** 2021-10-08

**Authors:** Carolina F. Rodrigues, Patrícia T. Borges, Magali F. Scocozza, Diogo Silva, André Taborda, Vânia Brissos, Carlos Frazão, Lígia O. Martins

**Affiliations:** 1Instituto de Tecnologia Química e Biológica António Xavier, Universidade Nova de Lisboa, Av da República, 2780-157 Oeiras, Portugal; carolinafrodrigues@itqb.unl.pt (C.F.R.); pborges@itqb.unl.pt (P.T.B.); darsilva@itqb.unl.pt (D.S.); ataborda@itqb.un.pt (A.T.); vbrissos@itqb.unl.pt (V.B.); frazao@itqb.unl.pt (C.F.); 2Instituto de Química Física de los Materiales, Medio Ambiente y Energia (INQUIMAE), CONICET—Universidad de Buenos Aires, Buenos Aires 148EHA, Argentina; magaliscocozza@gmail.com

**Keywords:** dye-decolorizing peroxidases, directed evolution, biorefineries, enzyme specificity, thermostability, structure–function relationships, soil bacteria

## Abstract

*Bacillus subtilis* *Bs*DyP belongs to class I of the dye-decolorizing peroxidase (DyP) family of enzymes and is an interesting biocatalyst due to its high redox potential, broad substrate spectrum and thermostability. This work reports the optimization of *Bs*DyP using directed evolution for improved oxidation of 2,6-dimethoxyphenol, a model lignin-derived phenolic. After three rounds of evolution, one variant was identified displaying 7-fold higher catalytic rates and higher production yields as compared to the wild-type enzyme. The analysis of X-ray structures of the wild type and the evolved variant showed that the heme pocket is delimited by three long conserved loop regions and a small α helix where, incidentally, the mutations were inserted in the course of evolution. One loop in the proximal side of the heme pocket becomes more flexible in the evolved variant and the size of the active site cavity is increased, as well as the width of its mouth, resulting in an enhanced exposure of the heme to solvent. These conformational changes have a positive functional role in facilitating electron transfer from the substrate to the enzyme. However, they concomitantly resulted in decreasing the enzyme’s overall stability by 2 kcal mol^−1^, indicating a trade-off between functionality and stability. Furthermore, the evolved variant exhibited slightly reduced thermal stability compared to the wild type. The obtained data indicate that understanding the role of loops close to the heme pocket in the catalysis and stability of DyPs is critical for the development of new and more powerful biocatalysts: loops can be modulated for tuning important DyP properties such as activity, specificity and stability.

## 1. Introduction

Biocatalysis is both a green and sustainable technology and redox biocatalysts offer eco-friendly advantages in comparison with conventional chemical reactions due to the selectivity, controllability and economy of their reactions. Lignin is the largest reserve of aromatics on Earth and is a key renewable source of chemicals and materials [1]. Recent strategies developed for lignin depolymerization allowed the derivation of well-defined compounds in acceptable quantities, bringing the utilization of lignin as a feedstock for aromatic chemicals one step closer to reality [1,2,3]. At present, the challenge is the set-up of atom-economic and waste-free (bio)processes that allow the full implementation of a lignin-derived platform of chemicals, sustainable starting material for the production of drop-in chemicals, structural scaffolds exploitable in the field of medicinal chemistry, polymers and emerging functional materials [1,4,5,6,7]. In nature, white-rot fungi and certain bacteria are responsible for the depolymerization and conversion of lignin, and therefore, they are useful sources of ligninolytic enzymes, such as laccases and fungal lignin (LiP), versatile (VP), manganese peroxidases (MnP) and dye-decolorizing peroxidases (DyPs) [8,9,10,11]. DyPs are a family of microbial heme peroxidases that display structural features analogous to chlorite dismutases with an α + β ferredoxin-like fold [12]. These enzymes show a remarkably broad range of substrates, from synthetic azo and antraquinonic dyes and aromatic sulfides to iron and manganese ions, phenolic and nonphenolic lignin units, wheat straw lignocellulose and kraft lignin, and are therefore interesting enzymes for a vast array of biotechnological applications including bioprocesses targeted at the valorization of lignin [9,13,14,15]. DyPs are classified into four distinct classes based on their primary structure, with A–C subfamilies of bacterial origin and the D subfamily of fungal origin [16]. An alternative tertiary structure-based classification subdivides DyPs into three classes: class I, intermediate, corresponds to former class A; class P, primitive, corresponds to former class B; and class V, advanced, containing members from former classes C and D [16]. Members from this class are in general the most efficient catalysts [16,17,18]. One aspect of interest in DyPs is that these enzymes are mostly present in bacteria and are then useful alternatives to fungal enzymes in lignin valorization, considering their, e.g., enlarged temperature and pH range [17]. Moreover, bacterial systems benefit from a wide range of molecular biology tools that facilitate their engineering towards improved stability, resistance to solvents and inhibitors and, thus, the set-up of sustainable processes by design. Recent reports on the engineering of DyPs have resulted in variants of *Pseudomonas putida Pp*DyP with improved catalytic efficacy for lignin-related phenolics, an upshift in the optimal pH to the alkaline range and enhanced resistance to H_2_O_2_ inactivation [19], representing excellent candidates for the development of H_2_O_2_ biosensors [20]. Moreover, the engineering of *Pseudomonas fluorescens* Dyp1B resulted in variants with increased *k*_cat_ for 2,6-dichlorophenol and the oxidation of alkali kraft lignin [21].

In a previous work, *Bs*DyP from the soil bacteria *Bacillus subtilis* was identified and characterized [22]. This is an interesting biocatalyst due to its high redox potential, broad substrate spectrum and very good thermal stability [22,23,24]. The *Bs*DyP enzymatic mechanism was the first among DyPs to be elucidated, revealing that the reduction of Compound II is the rate-limiting step of the catalytic cycle [25]. In spite of a broad substrate range, the activity of BsDyP for phenolics (*k*_cat_/Km ≈ 10^2^ M^−1^ s^−1^) is far from that observed for synthetic dyes (10^4^ M^−1^ s^−1^) and from that required for biotechnological applications. In the present work, *Bs*DyP was tailored to improve its catalytic efficiency for 2,6-dimethoxyphenol (DMP), a model lignin-derived phenolic, using directed evolution (DE) approaches. DE is a powerful engineering tool that mimics the principles of natural selection through iterative rounds of mutagenesis, recombination and screening [26]. The properties of evolved variants based on their biochemical, kinetic and structural analysis are discussed. In DyPs, the oxidation of reduced substrates occurs in heme cavities and in tyrosine and tryptophan surface-exposed residues, similarly to LiP and VP enzymes, and then to transfer electrons to the heme using long-range electron transfer (LRET) pathways [16,27,28]. However, details of substrate binding and of molecular determinants of substrate specificity in DyPs remain open questions. This work helps to understand the role of conserved loops around the heme pocket in substrate binding and catalysis and the interplay of catalytic and stability mechanisms of DyPs with implications in their industrial application and in the future design of enzymes.

## 2. Results and Discussion

### 2.1. Deletion of Tat-Signal Peptide from BsDyP

*Bs*DyP has a 45-residue N-terminal twin-arginine signal peptide sequence (MSDEQKKPEQIH**RR**DILKWGAMAGAAVAIGASGLGGLAPLVQTA) with an Arg-Arg motif, recognized by the twin-arginine translocation (Tat) pathway, involved in the translocation of folded proteins from the cytoplasm and secretion to the extracellular milieu [29,30]. The recombinant production of *Bs*DyP in *Escherichia coli* resulted in two forms with different molecular masses as assessed by SDS-PAGE (Appendix A [22]): one upper band that most probably corresponds to the unprocessed cytoplasmic precursor containing the signal peptide, and a lower band that corresponds to the mature periplasmic enzyme without the signal peptide [29], similarly to what was observed in the heterologous expression of TfuDyP *Thermobifida fusca* [31] and *E. coli* YcdB [32], DyP members from class I also harboring an N-terminal twin-arginine sequence. We deleted the N-terminal sequence of *Bs*DyP to achieve homogeneous preparations of recombinant enzyme. The heterologous production of the truncated (mature) form of *Bs*DyP resulted in one single band in the electrophoresis gel (Appendix A). The truncated enzyme shows spectroscopic and kinetic properties comparable to those of an intact enzyme [22]. This form (named hereafter wild-type *Bs*DyP) was used throughout this work.

### 2.2. Directed Evolution of BsDyP for Improved DMP Oxidation

Three rounds of evolution were performed and approximately 6000 clones were screened to identify a variant with improved catalytic efficiency for the lignin-related phenolic DMP (Figure 1). In the first round of evolution using epPCR, ~600 clones were screened using the qualitative “activity on-plate assay” followed by liquid activity screening in 96-well plates (Appendix A). Two variants, 3G5 (with mutations A330V, L166Q, V284A and T296S) and 1F9 (with the single S325P mutation), showed 2- and 5-fold higher activity for DMP, respectively, in comparison with the wild type (Appendix A). Variant 1F9 parented the next generation where ~3500 variants were screened (Appendix A). The top four variants inserted one additional mutation: K220R (variant 50F7), K317E (variant 54D6), E346V (variant 56C8), K248E (variant 51E9), and revealed very similar enzymatic activities (1.5 to 1.9-fold higher than 1F9), which impaired the selection of one clearly best variant to parent the following round of evolution (Appendix A). Therefore, to construct the third generation library of variants, random recombination by DNA shuffling of genes coding for 50F7, 51E9, 54D6 and 56C8 plus the gene coding for variant 3G5, found in the first round of evolution, was performed. A library of ~1980 mutants was screened and the top twenty-five variants were selected for further analysis (Appendix A). The DNA sequencing revealed that from the initial twenty-five phenotypes, only fifteen corresponded to different genotypes considering the non-synonymous substitutions (Appendix A); for example, the six top variants (5G5, 7E7, 6B8, 3C7, 2C5, 7E10) that reached 1.7 to 2.5-fold higher activity than the parent (considered 50F7) corresponded to only three genotypes. The highest activity for DMP was consistently measured for variant 5G5 (Appendix A), which gathered mutations S325P and A330V from variants 1F9 and 3G5, respectively, from the first generation, and mutation K317E from variant 54D6 of the second generation.

### 2.3. Biochemical and Kinetic Characterization

To understand the role of the three mutations that originated the improved activity of *Bs*DyP towards DMP, the wild type and variants 1F9, 3G5, 54D6 and 5G5 were overproduced, purified and characterized. The Reinheitszahl values of purified enzymes vary between 1.6 and 2.5 and the UV–visible absorption spectra of the purified enzymes revealed the characteristic Soret band at 406–407 nm (Table 1 and Appendix A). The enzyme yields improved in the course of evolution from 8 mg L^−1^ in the wild type to 15 mg L^−1^ in hit variant 5G5 (Table 1), indicating the beneficial role of mutations acquired for *Bs*DyP recombinant production. Furthermore, the incorporation of the heme cofactor almost doubled from 0.4 to 0.6–0.7 mol per mole of protein in the wild type and variants, respectively.

An optimal pH around 4 was observed for DMP and for 2,2′-azino-bis(3-ethylbenzothiazoline-6-sulfonic acid) (ABTS) for all tested enzymes (Appendix A). The activity for H_2_O_2_ and ABTS increased 2 to 4-fold in all variants, but as the K_m_ values also slightly increased (2–3 fold) the catalytic efficiency (*k*_cat_^app^/K_m_^app^) remained analogous in the variants as compared to the wild type (Table 2 and Appendix A). The variants from the first and second round of evolution, 1F9, 3G5 and 54D6, showed similar *k*_cat_^app^ values for DMP, circa 2-fold of those of the wild type (Table 3 and Appendix A). Noteworthy, the *k*_cat_^app^ of the hit 5G5 variant is ~7-fold higher for DMP, indicating a synergistic action of the three mutations S325P, A330V and K317E in the functional transition. The K_m_ values for DMP of variants 54D6 and 5G5 (both containing the mutation K317E) increased 10-fold as compared to the wild type, which is reflected in a slightly lower catalytic efficiency (*k*_cat_^app^/K_m_) for DMP as compared to the wild type. The lower Km indicates that mutation K317E introduced steric changes in the substrate-binding site(s) that negatively affected DMP binding to the enzyme. Note, however, that the turnover number (*k*_cat_^app^) is considered the most important parameter for biotechnological applications since bioprocesses usually take place at high concentrations of substrate, i.e., the enzyme’s activity is not limited by substrate concentration but by the turnover number. In this respect, the hit variant 5G5 has a *k*_cat_^app^ for DMP 107-, 56- and 29-fold higher when compared, for example, with class I bacterial *Thermobifida fusca Tfu*DyP (*k*_cat_^app^ = 0.026 s^−1^), class P *P. putida Pp*DyP (*k*_cat_^app^ = 0.05 s^−1^) and class V *Streptomyces avermitilis*, *Sa*DyP2 (*k*_cat_^app^ = 0.097 s^−1^), respectively [19,31,33]. 5G5 is also a promising candidate for further *Bs*DyP evolution to increase the *k*_cat_^app^ parameter to values close to those of fungal counterparts, such as *Auricularia auricula-judae*
*Aau*DyP, which shows a paramount *k*_cat_^app^ of 89 s^−1^ for DMP [34].

### 2.4. Role of Mutations in the Optimization of 5G5 for DMP Oxidation

To investigate the structural consequences of evolving the wild type to 5G5, the X-ray crystal structures of 5G5 and the wild type were solved at 2.10 and 2.49 Å resolution, respectively (Figure 2a and Table 4). The 5G5 variant crystal structure is very similar to the wild type, showing root-mean-square deviations (r.m.s.d.) between Cα atoms that range from 0.46 to 0.51 Å. The wild type crystallized in the trigonal P3121 space group with two molecules per asymmetric unit (Appendix A). These two subunits have an r.m.s.d. of 0.31 Å between Cα atoms. The recently deposited *Bs*DyP crystal structure (PDB 6KMN, [35]) belongs to the triclinic P1 space group, showing four subunits in the asymmetric unit. The crystal structure of the 5G5 variant belongs to the monoclinic P21 space group and four monomers in the asymmetric unit were identified. Chains A, B and C share a higher structural homology, with r.m.s.d. values ranging from 0.30–0.33 Å between Cα atoms, than chain D (r.m.s.d. = 0.43–0.45 Å); we opted to examine chain A for both the wild type and the 5G5 variant in most of the structural analysis.

The mutations present in 5G5 are located nearby the heme proximal side (Figure 2a). S325P is adjacent to the heme proximal histidine ligand, H326, and mutation A330V lines the cavity at ~3.5 Å to the heme. In the wild-type structure, the side chain of K317 is at the surface, located at ~10 Å from the heme propionate, but in 5G5, E317 is part of a region (309–321) that is not visible in the electron density maps (Figure 2b,c). When compared with the wild type, the 5G5 structure displays lower atomic displacement parameter (a.d.p.s) values, 68 Å^2^ and 38–48 Å^2^, respectively (Figure 2b,c and Table 4), but apparently the replacement of the positively charged K317 by the carboxylate glutamate resulted in the significantly higher flexibility of region 309–321 (see below).

In *Bs*DyP, the heme cofactor is partially buried in a predominantly hydrophobic pocket lined by the conserved proximal ligand H326 and catalytic distal residues D240 and R339 (Figure 3a,b and Appendix A). It is coordinated by four nitrogen atoms of the porphyrin ring, at a mean distance of 2.0 Å, and by H326 at 2.3 Å. The side chains of D240 and R339 are oriented to the heme iron, with a D240^OD1^–Fe distance of 5.2–5.5 Å and R339^NH2^–Fe distance of 4.1–4.6 Å, similar to those measured in the reported *Bs*DyP structure (PDB 6KMN). A water molecule is located approximately at 3.5 Å apart from the iron atom (Appendix A); in the heme pocket of the 6KMN structure, an electron density blob was fitted with molecular oxygen, where the closest oxygen atom to the iron atom is at a 3.4 Å distance [35]. Access to the heme from the surface, as defined using a 1.4-Å rolling probe, is made through a distal tunnel and an open wide cavity (Figure 3a and Appendix A). The tunnel, conserved in all characterized DyPs, is proposed to be the main entrance of H_2_O_2_ [36,37,38,39,40] and includes in *Bs*DyP a 6 Å-wide side gallery. The catalytic D240 and R339 residues are part of the tunnel, and three water molecules were found in this pathway in the wild-type structure (and four in the 5G5 structure). The open wide cavity gives access to the solvent-exposed heme propionate p6 group and likely represents an electron transfer route from substrates to the porphyrin radical [40,41]. *Bs*DyP structure PDB 6KMM unveiled the presence of two HEPES molecules: one is bound at 6 Å away from the propionate group in the heme cavity and another at 16 Å near the surface-exposed Y388 residue (Appendix A) [35]. Moreover, the docking of HEPES and three different synthetic dyes using snapshots from molecular dynamics simulations of *Bs*DyP also indicated two putative binding sites for reduced substrates [35]. *Bs*DyP, similarly to other DyPs, is rich in tyrosines and tryptophans (Appendix A) that can play a role in DyP catalysis by forming surface-exposed oxidation sites for bulky substrates, which are connected to the heme by long-range electron transfer (LRET) pathways [37,40,41,42,43,44,45,46,47,48]. Residue Y388 is the most exposed residue at the shortest distance to the heme and can hypothetically act as a second (radical) substrate-binding and -oxidation site in *Bs*DyP (Appendix A).

The mutations in the course of evolution of *Bs*DyP were inserted in flexible loops and in a small helix close to the heme pocket (Figure 3a,b). Flexible loops in the proximity of active sites frequently play roles in catalysis as they interact with solvent and substrates [53,54]; it is widely accepted that their pronounced conformational flexibility may contribute to their function, in substrate selectivity and recognition, and the facilitation of substrate binding, as well as protein evolution, as they represent molecular elements of variability [55]. Loop 1 (222–250) and loop 3 (334–342) are predominantly located at the distal side of the heme and comprise the catalytic residues D240 and R339, respectively. Loop 2 (297–325) and the small α-helix (326–333) are at the heme proximal side; loop 2 contains mutations K317E and S325P and the helix α6 comprises the adjacent heme proximal ligand H326 and the mutation A330V. A structural superposition of *Bs*DyP–HEPES bound (PDB 6KMM) with *Bs*DyP PDB 7PKX (this work) shows that a hydrogen bond network comprising solvent molecules and amino acids from loop 1, loop 2 and the small α6-helix (Appendix A) extended from HEPES to the heme propionate that may facilitate electron transfer to the heme iron. Therefore, the insertion of mutations in these flexible elements during the evolutionary trajectory of 5G5 is expected to have changed the conformational and chemical environment of the catalytic active site, facilitating redox reactions.

K317 (and D314 from loop 2) establishes salt bridges with D250 (and K248) from symmetry-related molecules in the wild-type crystal structure (Figure 4a). E312 from loop 2 interacts through hydrogen bonds with K239, T242, R299 and G304, which are at the cavity entrance (Figure 4a,c and Appendix A). Therefore, the replacement of K317 with a glutamate disrupted the salt bridge to D250, hypothetically leading to loop 2 destabilization and structural rearrangements that resulted in the higher flexibility of region 309–321 (Figure 4b). For example, R299 in 5G5 is at a different conformation and is closer to K331 (~5.0 Å) than in the wild type (~7.0 Å) and is significantly more exposed to the solvent, similarly to residues K239, T242 and G304 that border the entry to the cavity (Figure 4c,d and Appendix A). Furthermore, the higher flexibility of the region 309–321 in 5G5 allowed an increase in the solvent ASA of the residue P325 (Figure 4c,d). The tunnel that gives access to the heme has a similar length (~10 Å) and diameter (~3 Å) in both the wild type and the 5G5 variant (Appendix A), but the open cavity not only has a higher volume and area (Appendix A), but also an entrance width that is 6 Å larger in 5G5 (~18.0 Å) than in the wild type (~12.0 Å). This results in an increased exposure of the heme group to the solvent: 9% ASA in 5G5 as compared to 4% in the wild type (Figure 4c,d). The results indicate that the variation in the loops’ flexibility and the widening of the active site entrance in the evolved variant 5G5 negatively affect substrate binding, as reflected in higher Michaelis constants, K_m_, but simultaneously facilitate electron transfer from substrates to the heme, as assessed by the increased catalytic rates of 5G5 for DMP (and also for ABTS). The comparative analysis of available DyPs structures reveals that the loops and α-helix regions delimiting the heme pocket are structurally conserved in all DyPs (Figure 5). Members of class V display longer loops, on average with more than 10 to 30 residues, than enzymes from P and I classes. These longer loops most likely contribute to the occluding of the respective heme cavities from contact with the solvent, as observed in the X-ray crystal structures. Nevertheless, in solution, the expected high flexibility of this region may promote solvent access of the heme and allow the oxidation of small substrates at the cavity surface; for example, aromatic molecules were observed to bind within the heme entry of *Aau*DyP [27], and mutagenesis and kinetic analysis in this same enzyme indicated the involvement of a substrate-binding site close to the heme cavity [51].

### 2.5. Stability Properties of 5G5 as Compared to Wild Type

The thermodynamic stability of the wild type and variants was assessed by probing the unfolding of the tertiary structure using the fluorescence emission of the tryptophan residues (with selective excitation at 296 nm) in the presence of the chemical denaturant GdnHCl (Figure 6 and Appendix A). The wild-type *Bs*DyP displays a GdnHCl mid-point concentration (where 50% of molecules are unfolded) of 1.4 M, and the native state is more stable than the unfolded state by 7 kcal mol^−1^. The overall structure of the 5G5 variant is 2.0 kcal mol^−1^ less stable than the wild type and the GdnHCl mid-point decreases to 1.1 M. Variants 3G5 and 54D6 also displayed a lower overall stability similar to 5G5, while variant 1F9 was found to be comparable to the wild type (Appendix A).

The thermal robustness of *Bs*DyP and variants was also assessed. It was found that the insertion of mutations decreased the melting temperature (Tm, where 50% of the protein molecules are denatured) of all variants by 4–5 °C (Table 5 and Appendix A). Furthermore, the kinetic stability, which quantifies the amount of enzyme that loses activity irreversibly during incubation at a certain temperature, owing to misfolding, protein aggregation and covalent changes, reveals that the wild-type enzyme is clearly more stable than the variants with a 4-fold higher half-life at 40 °C (Table 4 and Appendix A). These data indicate unfavorable changes and an inverse relationship between activity and stability for the set of mutations present in the 5G5 variant.

Therefore, the insertion of mutations in BsDyP that enhanced catalytic rates was achieved at the expense of enzyme stability. An activity–stability trade-off is associated with improved function for phenolics in the course of the laboratory evolution of the *Bs*DyP enzyme. Similarly to the results obtained in this work, the improved activity towards lignin-derived phenolics of the enzyme variant *Pp*DyP-6E10 from *P. putida* MET94 also occurred at the expense of thermostability [19,20], and moreover, the three mutations inserted during the evolutionary trajectory of *Pp*DyP-6E10 were in the loop regions that surround the access to the heme cofactor of *Pp*DyP.

## 3. Materials and Methods

### 3.1. Bacterial Strains, Plasmids and Cultivation Media

*E. coli* strain DH5α (Novagen, Merck, Darmstadt, Germany) and *E. cloni* 10G (Lucigen, Middleton, WI, USA) were used for routine propagation and amplification of plasmid constructs. *E. coli* Tuner (DE3, Novagen, Merck, Darmstadt, Germany) and BL21 star (DE3, Novagen, Merck, Darmstadt, Germany) were used to heterologously express the *bsdyp* gene without signal peptide cloned in pET-21a (+) plasmid (Novagen, Merck, Darmstadt, Germany) (pVB4) and its variant genes. In *E. coli* BL21 star and *E. coli* Tuner, the target genes are under the control of the T7 promotor, induced by isopropyl β-D-1-thiogalactopyranoside (IPTG). Luria-Bertani medium (LB) was used as a routine liquid medium to grow the different *E. coli* strains, supplemented with appropriate antibiotics when required.

### 3.2. Construction of BsDyP Wild Type without Signal Peptide Sequence

The signal peptide present in the N-terminus of BsDyP was identified using Signal P [56] and removed using the plasmid pRC-2 [22] as a template and the primers *bsdyp*-A45M-NdeI-F (5′-GACTCATATGAAGCCATCGAAAAAG-3′) and *bsdyp*-BamHI-R (5′-CGCTGAAAGGATCCTGTAAAGGCTTCTTTTATGATTCC-3′), resulting in plasmid pVB4. The A45M mutation was introduced to create a new starting codon. PCR was carried out in a 50 μL containing 3 ng DNA template, 1 μM of primers, 200 μM of dNTPs, NZYProof polymerase buffer and 2.5 U of NZYProof polymerase (NZYTech, Lisboa, Portugal). After an initial denaturation period of 5 min at 94 °C, the following steps were repeated for 28 cycles in a thermal cycler (MyCyclerTM thermocycler, Biorad, Hercules, CA, USA): 1 min at 94 °C, 1 min at 57 °C, 2 min at 72 °C followed by a final 10 min period at 72 °C. The amplified products were purified using GFX PCR DNA and Gel Band Purification kit (GE Healthcare, Chicago, IL, USA). The final PCR product was digested with NdeI/BamHI (Thermofisher, Waltham, MA, USA), cloned into pET-21a (+) (Novagen, Merck, Darmstadt, Germany) and transformed into electrocompetent *E. coli* DH5α cells.

### 3.3. Random Mutagenesis by Error-Prone PCR and Mutant Library Construction

In the first round of evolution, the *bsdyp* wild-type gene was amplified using the forward primer *Bs*DyP-A45M-NdeI (5′-GACTCATATGAAGCCATCGAAAAAG-3′) and the reverse primer DyPBsRv (5′CGCTGAAAGGATCCTGTAAAGGCTTCTTTTATGATTCC3′). The epPCR was performed in a total volume of 50 µL reaction containing 3 ng of DNA template (pVB4), 1 µM of each primer, 200 µM of dNTPs, 4 mM MgCl_2_, Taq polymerase buffer and 2.5 U of Taq polymerase (Thermo Scientific, Waltham, MA, USA). The concentration of 0.01 mM MnCl_2_ was chosen for the construction of libraries of mutants. The PCR program was carried out in a thermal cycler (MyCyclerTM thermocycler, Biorad, Hercules, CA, USA) at the following conditions: 5 min initial denaturation at 94 °C, 28 cycles of 1 min at 94 °C, 1 min at 56 °C and 2 min at 72 °C and a final step of 10 min at 72 °C. The PCR product was purified using Illustra GFX PCR DNA kit (GE Healthcare, Chicago, IL, USA), digested with NdeI/BamHI and cloned into pET21a (+) (Novagen). Electrocompetent *E. coli* BL21 star cells were transformed with *Bs*DyP mutant libraries. The transformed cells were spread into LB solid medium supplemented with 100 µg mL^−1^ of ampicillin and 0.01 mM IPTG.

In the second round of evolution and in order to perform ligation through Gibson Assembly [57], the following primers were used to amplify *bsdyp* 1F9 gene: *Bs*DyPFwdGA (5′ CCACCAGTCATGCTAGCCATTTATGATTCCAGCAAACGC 3′) and *Bs*DyPRevGA (5′ TTTAAGAAGGAGATATACATATGAAGCCATCGAAAAAGG 3′). The underlined sequence is the region that will pair with the *bsdyp* gene template. The epPCR was performed in the presence of 0.1 and 0.15 mM MnCl_2_, and the annealing temperature was 59 °C. The amplification by PCR of the vector pET21a(+) was performed in high fidelity conditions with the forward primer pET21GA2F (5′ ATGTATATCTCCTTCTTAAAGTTAAACAAAATTATTTC 3′) and the reverse primer pET21GA240R (5′ ATGGCTAGCATGACTGGTG 3′). The PCR was performed in a final volume of 50 µL reaction containing 3 ng of DNA template (pET21a(+)), 0.5 µM of each primer, 200 µM of dNTPs, Q5 reaction buffer and 1 U of Q5 High-fidelity DNA polymerase (New England Biolabs). The PCR program was carried out in a thermal cycler (MyCyclerTM thermocycler, Biorad, Hercules, CA, USA) at the following conditions: 30 s initial denaturation at 98 °C, 30 cycles of 10 s at 98 °C, 20 min at 63 °C and 3 min at 72 °C and a final step of 10 min at 72 °C. The PCR products were purified using Illustra GFX PCR DNA kit (GE Healthcare, Chicago, IL, USA). To avoid the presence of the DNA template, both PCR products were digested with DpnI (Thermo Scientific, Waltham, MA, USA). The assembly of *bsdyp* variant genes to the expression vector pET21a(+) was performed using NEBuilder HiFi DNA Assembly Master Mix (NEB) and the mixture was used to transform E. cloni 10G electrocompetent cells (Lucigen). The colonies were washed with LB and the cells recovered by centrifugation at 4 °C, 3220× *g*, 20 min. The plasmids were extracted using GeneJET Plasmid Miniprep Kit (Thermo Scientific, Waltham, MA, USA). Electrocompetent *E. coli* BL21 star cells were transformed with *bsdyp* mutant libraries and were spread into LB solid medium supplemented with 100 µg mL^−1^ of ampicillin and 0.01 mM IPTG and plates were incubated overnight at 37 °C.

### 3.4. Recombination by DNA Shuffling and Mutant Library Construction

In the third round of evolution, DNA shuffling was performed after amplification of genes coding for variants 3G5, 50F7, 51E9, 54D6 and 56C8 using primers pET21GA2F (5′ ATGTATATCTCCTTCTTAAAGTTAAACAAAATTATTTC 3′) and pET21GA240R (5′ ATGGCTAGCATGACTGGTG 3′). A mixture containing 280 ng of each parental gene was digested with 0.15 U of DNase I in 200 mM Tris-HCl, pH 7, with 0.2 M MnCl_2_ for 20 min at 15 °C in a thermocycler. Digestion was stopped by adding 6 µL of 0.5 M EDTA. The PCR reassembly was carried out in a 20 µL reaction volume containing 5 µL of DNA fragments, 200 µM of dNTPs, NZYProof polymerase buffer and 2.5 U of NZYProof polymerase (NZYTech, Lisboa, Portugal). After an initial denaturation period of 3 min at 96 °C, the subsequent steps were repeated for 45 cycles in a thermal cycler: 1 min at 94 °C, 90 s at 59 °C, 90 s at 56 °C, 90 s at 53 °C, 90 s at 50 °C, 90 s at 47 °C, 90 s at 44 °C, 90 s at 41 °C and 1 min + 5 s/cycle at 72 °C followed by a final 10 min period at 72 °C. The PCR reassembly products were amplified by PCR using the primers *Bs*DyPFwdGA (5′ CCACCAGTCATGCTAGCCATTTATGATTCCAGCAAACGC 3′) and *Bs*DyPRevGA (5′ TTTAAGAAGGAGATATACATATGAAGCCATCGAAAAAGC 3′). The amplified products were purified using GFX PCR DNA and Gel Band Purification kit (GE Healthcare, Chicago, IL, USA). Ligation was performed through Gibson Assembly and the constructed library was firstly introduced into E. cloni 10 G electrocompetent cells, the plasmids were extracted and then introduced in *E. coli* BL21 star. The transformed cells were spread into LB solid medium supplemented with 100 µg mL^−1^ of ampicillin and 0.01 mM IPTG and plates were incubated overnight at 37 °C.

### 3.5. “Activity-On-Plate” Initial High-Throughput Screening

Colonies were replica-plated onto chromatography paper (Whatman, Maisdstone, UK) and the original culture-plate was re-incubated at 37 °C until colonies re-appeared (~6 h). The colonies on the filter papers were lysed by soaking the chromatography paper in a solution of 2 mg mL^−1^ of lysozyme (AppliChem, St. Louis, MO, USA) in 20 mM Tris-HCl, pH 7.6, and the papers were placed at 37 °C for 2 h. After the incubation, the chromatographic papers were soaked in the reaction mixture containing 100 mM sodium acetate buffer, pH 3.8, 1 mM DMP and 0.2 mM H_2_O_2_. After this step, the appearance of an orange color at room temperature indicated the presence of enzymatic activity in colonies.

### 3.6. Activity Screening in 96-Well Plates

Colonies that screened positive using the “activity-on-plate” assay were picked from the original culture-plate and transferred to 96-well plates containing 200 µL of LB supplemented with 100 µg mL^−1^ of ampicillin. Four wells in each plate were used to inoculate the parental strain of each generation as a control. Plates were incubated for 6 h at 37 °C, 750 rpm in Titramax 1000-plate shaker (Heidolph, Schwabach, Germany), after which 0.01 mM IPTG and 15 µM Hemin were added. Cultures were cultivated overnight and cells were harvested by centrifugation (5 min at 3220× *g*, 4 °C) and disrupted using 3 cycles of freeze and thaw, and pellets were suspended in 100 µL of 20 mM Tris-HCl buffer, pH 7.6, and lysozyme (2 mg mL^−1^). Plates were centrifuged (30 min at 3220× *g*, 4 °C) and the supernatants were collected (cell crude extracts) and used for enzymatic activity.

To test the enzymatic activity of variant libraries, 20 µL of cell crude extracts was transferred to 96-well plates containing 180 µL of reaction mixture (100 mM sodium acetate, pH 3.8, 1 mM DMP and 0.2 mM H_2_O_2_). The progress of reactions was monitored at 468 nm (ε_498_ = 49,600 M^−1^ cm^−1^) using a Synergy2 microplate reader (BioTek, Winooski, VT, USA). One unit (U) of enzymatic activity was defined as the amount of enzyme required to reduce 1 µmol of the substrate per minute.

### 3.7. Activity Re-Screenings at Large Scale

Variants showing higher activity than the parental strain were rescreened to eliminate false positives. Recombinant strains were cultivated in 50 mL of LB medium, and 0.1 mM IPTG and 75 µM hemin were added when OD_600_ = 0.6. Cells were harvested after 24 h and disrupted using a French press [22]. The protein concentration in cell crude extracts was measured using the Bradford assay. Enzymatic activity for DMP was performed as described above. In each generation, the variant with the highest activity was chosen to be the parent of the following generation. DNA sequencing using T7 terminator universal primers identified the mutations introduced.

### 3.8. Production and Purification of Wild-Type BsDyP and Variants

The plasmid pVB4, containing the wild-type *bsdyp* gene without a signal peptide, and plasmids harboring the hit variant genes were introduced into *E. coli* Tuner (DE3, Novagen, Merck, Darmstadt, Germany), which were cultivated in 2.5 L of LB medium. Crude extracts were loaded onto a SP-Sepharose column (GE, Healthcare, Bio-Sciences, Chicago, IL, USA), previously equilibrated with 20 mM Tris-HCl buffer, pH 7.6. Elution was performed by applying a 0–50% gradient with 1 M NaCl in 5 column volumes, followed by a second gradient of 50–100% in 2 column volumes. The active fractions were collected, pooled and concentrated, before applying onto a Superdex 75 HR 16/60 (GE Healthcare, Bio-Sciences, Chicago, IL, USA), pre-equilibrated with 20 mM Tris-HCl buffer with 0.2 M NaCl, pH 7.6 [22]. The protein concentration was determined using the extinction coefficient ε_280_ = 36,900 M^−1^cm^−1^. The heme content was determined by the pyridine ferrohemochrome method using an extinction coefficient of Ɛ_R-O 556_ (28.32 mM^−1^cm^−1^) [58]. UV–visible absorption spectra of purified enzymes were performed in a Nicolet Evolution 300 spectrophotometer (Thermo Industries, Waltham, MA, USA).

### 3.9. Apparent Steady-State Kinetic Analysis

The pH profiles of the wild type and variants were determined using 1 mM substrate (ABTS or DMP) in the presence of 0.2 mM of H_2_O_2_ in Britton–Robinson buffer (100 mM phosphoric acid, 100 mM boric acid and 100 mM acetic acid mixed with 1 M of NaOH to the desired pH in the range 2–10 at 25 °C. The enzymatic activities for DMP (0.001–6 mM, pH 3.8) and ABTS (0.01–5 mM, pH 4.2) were performed in the presence of 0.2 (wild type) and 0.6 mM H_2_O_2_ (variants 1F9, 3G5, 54D6 and 5G5). The rates varying the concentration of H_2_O_2_ (0.01–6 mM) were measured in the presence of 4 mM ABTS for the wild type and variants 3G5, 54D6 and 5G5, and 1 mM ABTS for variant 1F9. The enzymatic rates were calculated considering the active fraction of enzyme preparations (i.e., considering the holoprotein concentration based on the ratio heme:protein). The apparent steady-state kinetic parameters were determined by fitting data directly into the Michaelis–Menten equation (Origin95^®^software, OriginLab, Northampton, MA, USA). All assays were performed at least in triplicate.

### 3.10. Crystallization

Wild-type *Bs*DyP crystallization screenings, Structure Screen I and II and JCSG-plus (Molecular Dimensions, Sheffield, UK), were performed at 20 °C using the vapor diffusion technique and a Mosquito nano-drops dispenser robot (TTP Labtech, Melbourn, UK) to simultaneously test 1:1, 1:2 and 2:1 relative volumes of protein and crystallization solution droplets. Needle-shaped orange crystals appeared within three days in 0.05 M potassium phosphate monobasic and 20% (*w*/*v*) PEG 8K in a 2:1 protein:crystallization volume ratio. Crystals were optimized using manual handling at micro-liter scale. The best native crystal was obtained in 0.05 M potassium phosphate monobasic and 15% (*w*/*v*) PEG 8K. Crystals of *Bs*DyP 5G5 variant were produced in 24 h upon mixing 1 µL reservoir solution with 0.1 M Tris-HCl, pH 8.5, 25% (*w*/*v*) PEG 4K, 0.2 M magnesium chloride and 2 µL of protein solution. Native and 5G5 variant crystals were soaked in cryo-protectant solutions, each composed of the corresponding crystallization solution supplemented with 20% (*v*/*v*) glycerol prior to flash-cooling to 100 K.

### 3.11. Data Collection

Wild-type X-ray diffraction data were collected at 100 K at ALBA synchrotron (Barcelona, Spain) on the BL13-XALOC beamline equipped with a PILATUS 6M detector and an MD2M diffractometer [59]. 5G5 variant diffraction data were measured at the microfocus beamline ID30A-3 of the European Synchrotron Radiation Facility (ESRF, Grenoble, France), Grenoble, France using an Eiger X 4M detector and a MD2 micro-diffractometer [60]. XDS [61] was used for diffraction spot indexing, integration, scaling and merging into the final amplitudes dataset. Wild-type and 5G5 variant data were processed in space group P3121 and P1211, respectively. Data collection details and respective processing statistics are listed in Table 4.

### 3.12. Structure Determination and Refinement

The solvent contents of each crystal were estimated using the corresponding Matthews coefficients [62,63]. The phase problems were solved by molecular replacement. Wild-type *Bs*DyP structure was solved using MORDA [64] and the structure coordinates of *Streptomyces coelicolor* DyP (PDB 4GRC) as a search model, while the 5G5 variant structure was solved using PHASER [65], within the PHENIX suite [66,67], with the coordinates of the native crystal as a search model. The solution of the native structure by MORDA specified 0.74 Q-factor, 0.41 Rfree value and a probability of 0.99. The solution of the 5G5 variant obtained by PHASER showed TFZ values of 15, 34, 43 and 92, which successfully validate the solution [68].

Both structures were refined with PHENIX.REFINE (1.19-4122, Berkeley, USA) [66,67,69,70] using 1.5% of random reflections in thin-resolution shells for Rfree monitoring and to steer the relative weights of stereochemical restraints versus experimental data of the minimization function. The TLSMD server (http://skuld.bmsc.washington.edu/~tlsmd, accessed on 6 January 2020) [71] was used to define structural regions of translation, libration and screw (TLS) refinement of atomic displacement parameters. Refinement included atomic coordinates, individual isotropic atomic displacement parameters (*a.d.p.s*), TLS refinement and automatic water solvent completion with hydrogen bonding distances within 2.45–3.40 Å. Although the refinement included standard stereochemistry libraries [72], inter-atomic distances involving iron–heme cofactors were refined without target restraints.

Structure refinement was performed in iteratively repeated cycles of protein and solvent updating and refinement, alternated with inspection of σA-weighted 2|F_o_|−|F_c_| and |F_o_|−|F_c_| electron density maps for manual model improvement with COOT [73]. MOLPROBITY [74] was used to validate the model stereochemistry. Three-dimensional superposition of polypeptide chains was performed with MODELLER [75]. The analysis of molecular tunnels was performed with CAVER [76] and PyMOL [77,78]. Cavities were determined using Dogsitescorer program (2012, Hamburg, Germany) [79]. ASA was calculated with AREAIMOL [80,81]. Figures of structural models were prepared with PyMOL [77,78]. Refinement statistics are presented in Table 4. Structure factors and associated structure coordinates of wild-type *Bs*DyP and 5G5 variant were deposited in the Protein Data Bank (www.rcsb.org, accessed on 27 August 2021) [82] with PDB codes 7PKX and 7PL0, respectively.

### 3.13. Enzyme Stability

Thermal denaturation was assessed by steady-state fluorescence measured with a Cary Eclipse Spectrofluorometer (Agilent Technologies, Santa Clara, CA, USA) at excitation wavelength of 296 nm and emission wavelength of 330 nm [22]. Preparations containing 0.2 mg mL^−1^ protein in 20 mM sodium acetate, pH 5, were placed onto a thermostatically controlled thermal block and heated at a rate of 1 °C/min up to 90 °C. For the equilibrium unfolding studies, guanidinium hydrochloride (GdnHCl) concentrations in the range of 0–2.5 M in 20 mM Tris–HCl containing 0.2 M NaCl buffer, pH 7.6, were used to induce protein unfolding. The excitation wavelength was 296 nm and the fluorescence emission was recorded between 310 and 450 nm. The stability of enzymes was analyzed based on a two-state process [83]. Thermal inactivation assays (kinetic stability) were performed as previously described [22]. Briefly, enzyme preparations were incubated at 40 °C in 20 mM Tris-HCl buffer, pH 7.6, containing 0.2 M NaCl, and at time intervals, samples were withdrawn and tested for activity. Inactivation constants *k*_in_ were obtained by linear regression of (ln activity) versus t. Thermal inactivation appears to obey first-order kinetics, and the half-life value (t_1/2_) was calculated using t_1/2_ = ln2/*k*_in_. All assays were performed at least in triplicate.

## 4. Conclusions

DyPs are potentially important biocatalysts for industrial oxidations and redox conversions of lignin-related phenolics. In this study, *Bs*DyP was evolved through rounds of epPCR and DNA shuffling followed by high-throughput screening. This led to the identification of the variant 5G5, showing almost 10-fold higher activity for a lignin-related phenolic and improved enzyme production yields. Furthermore, the biochemical and kinetic analysis of intermediate variants and the comparative examination of the X-ray crystal structure of the wild type and variant 5G5 revealed novel insights into the structure–function relationship contributing to advancing the knowledge of enzyme function and evolution in DyPs. It was evidenced that loops around the heme pocket in DyPs account for a local flexibility, which is critical for modulating important enzyme properties such as activity, specificity and stability. The obtained results suggest that it is possible to tune the dynamics of catalytically relevant loops, which might be essential for the improvement or emergence of new catalytic properties in these enzymes. This work opens perspectives to further evolve BsDyP towards the conversion of phenolic compounds in green chemistry and biorefinery fields and to advance fundamental biochemical insights within the DyP-type peroxidase family of enzymes.

## Figures and Tables

**Figure 1 ijms-22-10862-f001:**
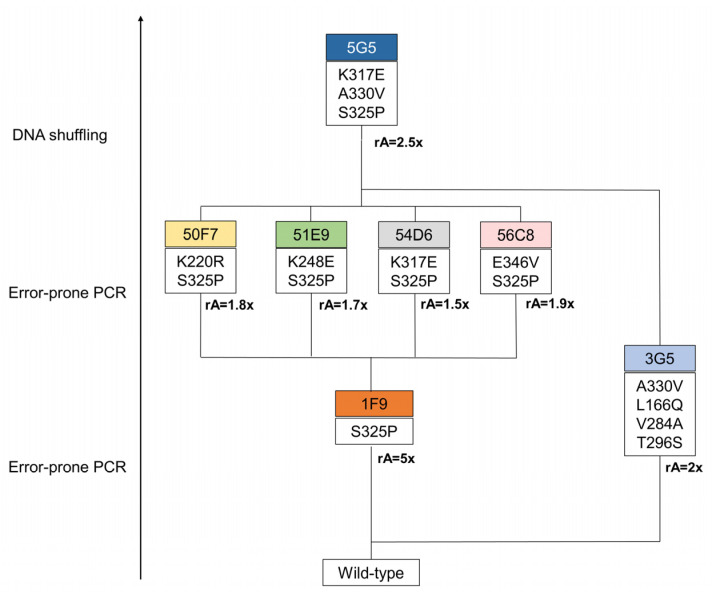
Lineage of *Bs*DyP variants generated in this study. In the first generation, a total of 614 variants, generated through error-prone PCR, were screened in 96-well plates using 1 mM of DMP. The 1F9 variant was selected on the basis of its 5-fold increased activity in crude extracts compared with the parent. Next, the second generation was evolved from the 1F9 variant as the parent. In this round, 365 variants, resulting from error-prone PCR, were screened in 96-well plates. The variants 50F7, 51E9, 54D6 and 56C8 were identified as having a slightly higher activity (between 1.5 and 1.9-fold) when compared to the parent. In the third generation, variants were constructed using DNA shuffling, where genes from variants of the second generation were recombined with the 3G5 gene, from the first round of evolution, with a 2-fold increased activity in crude extracts when compared with the wild type. In this round of evolution, 588 variants were screened in 96-well plates. From this process resulted one hit variant, 5G5, with 2-fold increased activity at pH 4. rA represents the relative activity to the parent.

**Figure 2 ijms-22-10862-f002:**
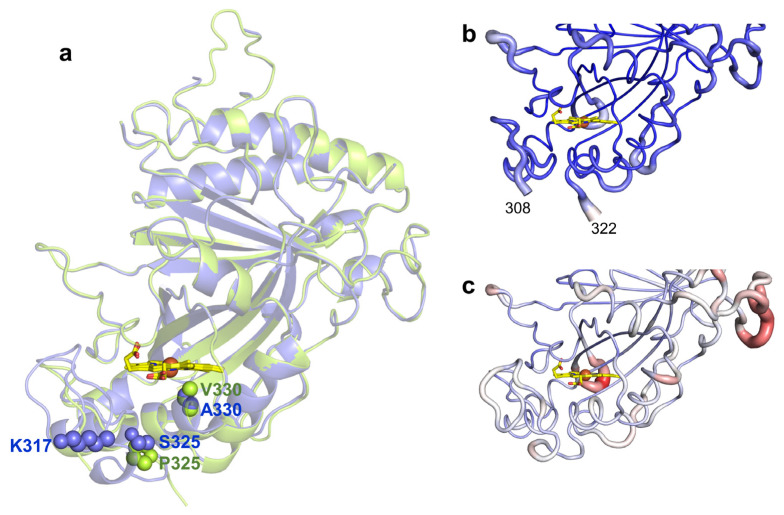
(**a**) Transparent secondary structure cartoon of superimposed wild type (blue) and 5G5 (green). The mutations P325 and V330 in 5G5 are shown as green spheres, and their homologs in the wild type as blue spheres. K317 is shown as blue spheres in the wild type and E317 is not visible in 5G5, and therefore is not represented. The heme is represented as sticks, with carbon, oxygen and nitrogen atoms colored as yellow, red and blue, respectively. The iron is shown as a brown sphere. Cartoon representation of the main-chain 5G5 (**b**) and wild-type (**c**) structures with thickness proportional to a.d.p. values, color coded from blue (21 Å^2^) to red (126 Å^2^). The full length enzymes contain 416 residues but the refined models consist of polypeptide chains with 349–354 amino acids; the first 45 residues, TAT signal, were deleted and the residues 46–55 (N-terminal) and 415–416 (C-terminal) are not visible in the electron density maps. A long loop region containing 32 residues (107–138) has high a.d.p.s, 55.4–109.8 Å^2^, and five residues (112–116) could not be modeled in the wild type due to lack of electron density. Additionally, the region 309–321 (boxed) becomes more flexible during evolution and is not visible in the electron density maps of 5G5 in chains A–C (**b**); in chain D, the missing region is between residues 317 and 321.

**Figure 3 ijms-22-10862-f003:**
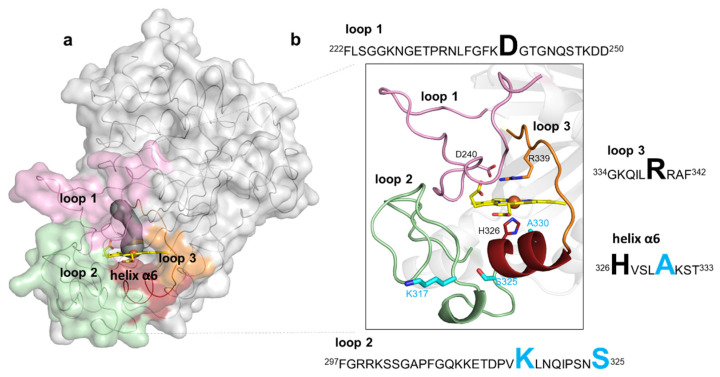
Representation of loops and the small helix delimiting the heme pocket in *Bs*DyP. (**a**) The *Bs*DyP monomeric form is shown as solvent-accessible surface colored in gray. The loops that surround the heme pocket are shown in pink (loop 1), green (loop 2) and orange (loop 3) and a small helix α6 is represented in dark red. The access to the heme, as defined using a 1.4-Å rolling probe, is made through a distal tunnel (in dark gray) and one cavity; (**b**) Zoomed view of the cartoon representation as shown in (**a**). The catalytic residues, D240 and R339, are shown as sticks with carbon atoms colored in pink and orange, respectively. The heme proximal ligand H326 is shown as sticks with carbon atoms colored in red. The residues K317, S325 and A330 that were replaced in the evolved variant 5G5 are shown as sticks with carbon atoms colored in cyan. The nitrogen and oxygen atoms are shown as sticks colored in blue and red, respectively. Highlighted in the text boxes are the catalytic residues (D240 and R339) and the heme proximal ligand (H326) in black, and the residues (K317, S325 and A330) that were replaced during the course of evolution in light blue.

**Figure 4 ijms-22-10862-f004:**
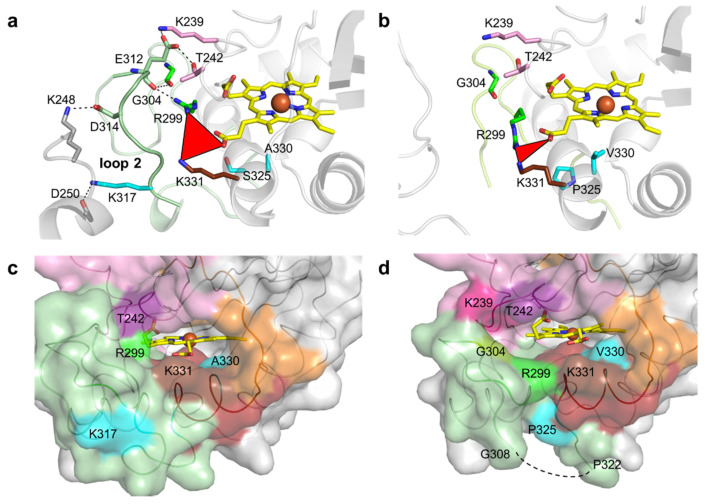
Residues surrounding the heme cavity in *Bs*DyP wild type and 5G5. Stick representation of residues limiting the access to heme cavity and interacting with residues K317, D314 and E312 of loop 2 in wild type (**a**) and 5G5 variant (**b**). The carbon atoms of the mutated residues (S325P, A330V and K317E) are colored in cyan. The residues K239 and T242 (loop 1) and R299 and G304 (loop 2) are interacting through hydrogen bonds with E312 (loop 2) and are colored accordingly (Figure 3). The residues of the symmetry-related molecule K248 and D250 are represented with carbon atoms colored in gray. The hydrogen bonds between residues are shown as dashed lines. The red triangle represents the interatomic distances between R299, K331 and the heme propionate group. Solvent-accessible surface area (ASA) representation of the regions delimiting the cavity is colored as in Figure 3 for wild type (**c**) and 5G5 (**d**). The residues with higher ASA values (K239, T242, R299, G304 and K331) in 5G5 are highlighted. The mutated residues K317, P325 and A/V330 are colored as in (**a**) and (**b**). The missing region 309–321 is shown as a dashed line in (**d**).

**Figure 5 ijms-22-10862-f005:**
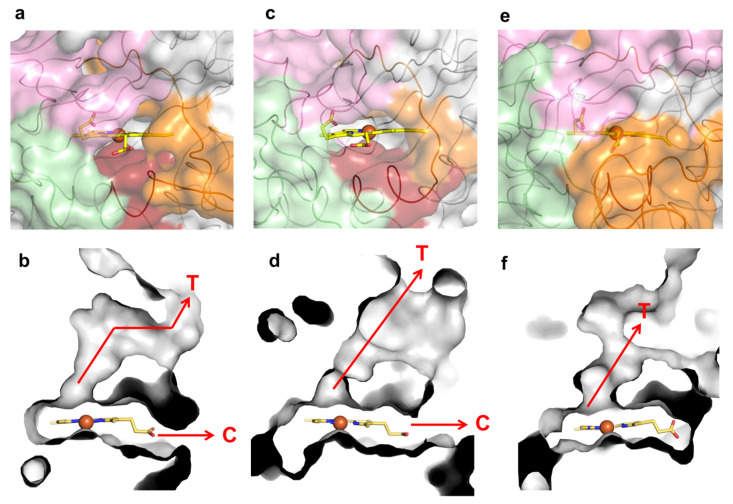
Loops, molecular tunnels and cavities in class P *K. pneumoniae* DyP (PDB 6FKS) (**a**,**b**), class I *B. subtilis Bs*DyP (PDB 7PKX) (**c**,**d**) and class V *A. auricula-judae* DyP (PDB 4AU9) (**e**,**f**). The regions lining the cavities are colored in light pink (loop 1), green (loop 2), dark red (small α-helix) and orange (loop 3). The tunnels (T) and cavities (C) corresponding to panels (**a**,**c**,**e**) are represented as grey ASA in panels (**b**,**d**,**f**), respectively.

**Figure 6 ijms-22-10862-f006:**
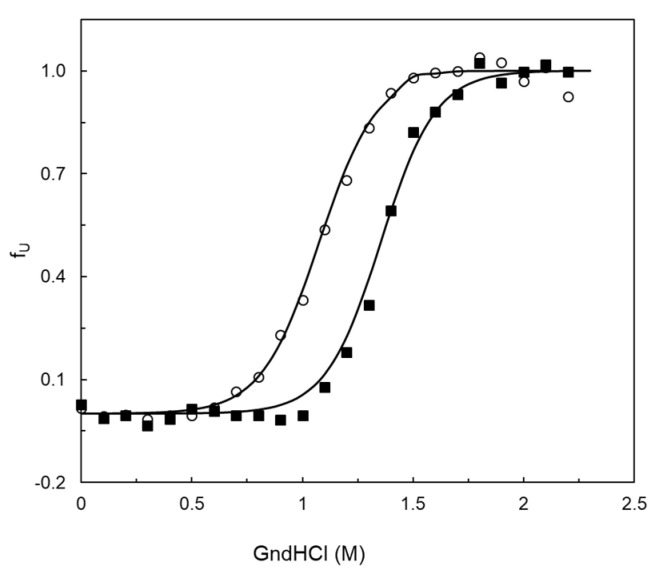
Unfolded fraction (fU) of wild-type *Bs*DyP (closed squares) and 5G5 (open circles) using guanidine chloride (GdnHCl) at pH 7.5. The solid lines are the fit according to the equation fU = exp(−ΔG°/RT)/1 + exp(−ΔG°/RT), which assumes the equilibrium N ⇄ U.

**Table 1 ijms-22-10862-t001:** Spectroscopic and redox properties of purified *Bs*DyP wild type and variants.

Enzyme	Production (mg L^−1^)	Rz	λmax (nm)	ε (mM^−1^ cm^−1^)	Heme Content
Wild Type	8.3 ± 0.4	2.3	406	82	0.4 ± 0.1
1F9	11.1 ± 0.8	2.5	407	92	0.7 ± 0.2
3G5	6.5 ± 0.5	1.6	406	60	0.6 ± 0.2
54D6	16.0 ± 0.9	1.8	407	66	0.6 ± 0.1
5G5	15.0 ± 1.0	2.1	407	77	0.7 ± 0.1

**Table 2 ijms-22-10862-t002:** Apparent steady-state kinetic parameters of wild type and variants for hydrogen peroxide and ABTS, measured at 25 °C and pH 4.2.

Enzyme		H_2_O_2_			ABTS	
	*k*_cat_^app^ (s^−1^)	K_m_^app^ (mM)	*k*_cat_^app^/*K_m_*^app^ (M^−1^s^−1^)	K_i_ (mM)	K_m_^app^ (mM)	*k*_cat_^app^/*K_m_*^app^ (M^−1^s^−1^)
Wild Type	14.2 ± 1.0	0.06 ± 0.01	(2.2 ± 0.4) × 10^5^	6.5 ± 1.9	0.42 ± 0.03	(3.6 ± 0.3) × 10^4^
1F9	50.4 ± 1.4	0.19 ± 0.05	(2.7 ± 0.7) × 10^5^	2.6 ± 0.2	0.89 ± 0.10	(5.8 ± 0.7) × 10^4^
3G5	28.1 ± 2.1	0.06 ± 0.01	(4.9 ± 0.7) × 10^5^	8.5 ± 0.4	1.37 ± 0.25	(2.5 ± 0.5) × 10^4^
54D6	47.7 ± 2.9	0.15 ± 0.02	(3.3 ± 0.6) × 10^5^	8.1 ± 0.7	0.80 ± 0.14	(6.0 ± 1.1) × 10^4^
5G5	56.1 ± 1.9	0.18 ± 0.02	(3.3 ± 0.7) × 10^5^	3.9 ± 0.6	1.29 ± 0.19	(4.2 ± 0.6) × 10^4^

**Table 3 ijms-22-10862-t003:** Apparent steady-state kinetic parameters of wild type and variants for DMP, measured at 25 °C in sodium phosphate buffer at pH 3.8 in the presence of 0.2 (wild type) and 0.6 mM (1F9, 3G5, 54D6 and 5G5) H_2_O_2_.

Enzyme	*k*_cat_^app^ (s^−1^)	*K_m_*^app^ (mM)	*k*_cat_^app^/*K_m_*^app^ (M^−1^s^−1^)
Wild Type	0.42 ± 0.03	0.06 ± 0.01	(7 ± 1) × 10^3^
1F9	0.83 ± 0.04	0.15 ± 0.01	(5.5 ± 0.4) × 10^3^
3G5	0.8 ± 0.1	0.09 ± 0.01	(9 ± 1) × 10^3^
54D6	1.0 ± 0.1	0.60 ± 0.01	(1.6 ± 0.5) × 10^3^
5G5	2.8 ± 0.1	0.7 ± 0.1	(3.8 ± 0.2) × 10^3^

**Table 4 ijms-22-10862-t004:** X-ray data collection and refinement statistics. Values in parentheses belong to the highest resolution shell.

	*Bs*DyP-Wild-Type	*Bs*DyP-5G5
**Data Collection**		
Beamline	BL13-XALOC	ID30A-3
Wavelength (Å)	0.97926	0.9680
Space group	*P 3_1_ 2 1*	*P 1 2_1_ 1*
Unit cell parameters (Å)	a = 95.3, b = 95.3, c = 181.2	a = 63.9, b = 114.8, c = 116.8
Resolution (Å)	75.09–2.49 (2.59–2.49)	61.96–2.10 (2.20–2.10)
Number of observations	215,157 (26,986)	285,233 (46,534)
Unique reflections	33,999 (5374)	92,499 (15,038)
Completeness (%)	99.8 (99.2)	99.1 (98.0)
Multiplicity	6.3 (5.0)	3.1 (3.1)
Mosaicity (ᵒ)	0.12	0.09
CC_1/2_ (%) ^a^	99.7 (30.6)	99.6 (47.8)
R_sym_ (%) ^b^	13.0 (78.8)	8.8 (53.2)
R_meas_ (%) ^c^	16.8 (243.4)	12.2 (111.2)
R_pim_ (%) ^d^	5.6 (39.5)	5.8 (35.2)
<I/σ(I)>	10.37 (0.67)	9.02 (1.39)
Wilson B-factor (Å^2^)	61.4	42.4
V_M_ (Å^3^ Da^−1^)	2.99	2.27
Estimated solvent content (%)	58.9	45.8
**Refinement**		
R_work_ (%) ^e^	20.7	19.1
R_free_ (%) ^e^	23.2	21.5
rmsd for bond lengths (Å)	0.008	0.002
rmsd for bond angles (°)	0.922	0.627
Average B-factor (Å^2^)	68.3 (chain A), 68.2 (chain B)	37.8 (chain A), 39.5 (chain B), 43.2 (chain C), 47.6 (chain D)
Ramachandran plot		
Residues in favored regions (%)	97.6	97.4
Residues in allowed regions (%)	2.4	2.6
Residues in disallowed regions (%)	0	0
PDB code	7PKX	7PL0

^a^ CC 1/2 = Percentage of correlation between intensities from random half-datasets [49]. ^b^ Rsym = Σhkl Σi | Ii(hkl) − <I(hkl)> |/Σhkl Σi Ii (hkl), where Ii(hkl) is the observed intensity and <I(hkl)> is the average intensity of multiple observations from symmetry-related reflections [50]. ^c^ Rmeas = Σhkl [N/(N(hkl) − 1)]1/2 Σi | Ii(hkl) − <I(hkl) >|/Σhkl Σi Ii (hkl), where N(hkl) is the data multiplicity, Ii(hkl) is the observed intensity and <I(hkl)> is the average intensity of multiple observations from symmetry-related reflections. It is an indicator of the agreement between symmetry-related observations [51]. ^d^ Rp.i.m. = Σhkl [1/(N(hkl) − 1)]1/2 Σi | Ii(hkl) − <I(hkl) >|/Σhkl Σi Ii (hkl), where N(hkl) is the data multiplicity, Ii(hkl) is the observed intensity and <I(hkl)> is the average intensity of multiple observations from symmetry-related reflections. It is an indicator of the precision of the final merged and averaged dataset [52]. ^e^ Rwork refers to the actual working dataset used in refinement, while Rfree refers to a cross validation set that is not directly used in refinement and is therefore free from refinement bias.

**Table 5 ijms-22-10862-t005:** Thermal stability of wild-type and variant enzymes as monitored by fluorescence emission (tertiary structure) at 330 nm and thermal inactivation at 40 °C. T_m_ is the melting temperature, where 50% of the protein molecules are unfolded. Half-life at 40 °C is the time after which 50% of activity is achieved. Assays were performed in triplicate.

Enzyme	T_m_ (°C)	Half-Life 40 °C (h)
WT	62.4 ± 0.7	109 ± 1
1F9	59.2 ± 0.4	21 ± 1
3G5	58.1 ± 0.2	27 ± 3
54D6	58.2 ± 0.4	10 ± 2
5G5	58.6 ± 0.4	20 ± 4

## Data Availability

The raw data supporting the conclusions of this article will be made available by the authors, without undue reservation, or are already included in the main manuscript and its Appendix A.

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
