# Peer review of "Loops around the Heme Pocket Have a Critical Role in the Function and Stability of BsDyP from Bacillus subtilis"

_ijms, 2021, doi:10.3390/ijms221910862_

Round 1

Reviewer 1 Report

The manuscript contains new and satisfactory information about of structural characterization of the produced BsDyP, and BsDyP enzyme characterization regearding to the enzyme stability, enzyme kinetics with two different subtrates (ABTS and DMT) and enzyme thermal properties. All mentioned analyses and results very well justify manuscript publication.

The reviewer recommends below: Recommending acceptance with minor revision. 

Reviewer comment:

The present research manuscript is introduced in a form of original scientific paper. Presented manuscript, from the aspect of the subject, is suitable for publication in International Journal of Molecular Sciences. This study investigate the strategies based on the optimization of dye-decolorizing peroxidases from Bacillus subtilis (BsDyP) using directed
evolution for improved oxidation of 2,6-dimethoxyphenol, a model lignin-derived phenolic. After three rounds of evolution, one variant was identified displaying 7-fold higher catalytic rates and higher production yields as compared to the wild-type enzyme. In short, BsDyP from the soil bacteria Bacillus subtilis was identified and characterized previously, and now it was tailored in order to improve catalytic efficiency for 2,6-dimethoxyphenol (DMP), a model lignin-derived phenolic, using directed evolution (DE) approaches. Special viewpoint was analysis of BsDyP Xray structures. The structure analysis show that the heme pocket is delimited by three long conserved loop regions and a small α-helix where, incidentally, the mutations were inserted in the course of evolution. One loop in the proximal side of the heme pocket becomes more flexible in the evolved variant, the size of active site cavity is increased, as well as the width of its mouth, resulting in an enhanced exposure of the heme to solvent. These conformational changes have a positive functional role in facilitating electron transfer from the substrate to the enzyme.

The manuscript contains new and satisfactory information about of structural characterization of the produced BsDyP, and BsDyP enzyme characterization regearding to the enzyme stability, enzyme kinetics with two different subtrates (ABTS and DMT) and enzyme thermal properties. All mentioned analyses and results very well justify manuscript publication.

The quality of cited literature data and presented results in the manuscript is satisfied. The
paper is scientifically sound, it is proper written and well organized, and it can be understood by
non-specialists.

According mentioned above information, the reviewer recommends below: Recommending
acceptance with minor revision. The partial sections in the text of manuscript and in the results
have to be clarified or fixed/reorganized which could improve the quality of manuscript: 

#1 – My suggestion is to add one more key words due to the topic, which will emphasize the
strain Bacillus subtilis, and thus improve the visibility of the scientific work

#2 – Abstract is generally appropriate, but it can be improved if the authors write all sentences
in the third person singular/ plural, not in the first. So, avoid telling “we, ours”, already say “in
this paper, this research, showed results”, etc. Carefully take into account pronouns in the entire
text of the paper and use pronouns in the third person.

#3 – Graphical abstract (GA) is desirable; it seems that the scope of research is very interesting, so the presentation of well organized and systematic graphical abstract is credible way to upgrade the quality of this manuscript. The authors are suggested to look at the instructions of the journal and to prepare the GA in accordance with it.

Answer: We have prepared and submitted a graphical abstract, the reasons the reviewers did not access it is not clear to us.

#4 – The section Introduction is necessary to modify, correct ithe initial paragraph in which
author describe the lignocellulolitic biomass, and its composition. For this type od manuscript, this
is unappropriate, because the authors didn’t use lignocellulose for cultivation of bacteria strain, or
for fermentation in the proces of peroxidase production. So, first paragraph isunnecessary. Second
paragraph may be the first: “Biocatalysis is both a green and sustainable technology and redox
biocatalysts offer…”. Here, the authors can state how important oxidative degradation of lignin is
and state which are the two major groups of enzymes used (heme peroxidases and laccases). Than,
the clasification of lignin-degrading peroxidases which include
❖ lignin peroxidase (LiP);
❖ manganese-dependent peroxidase (MnP),
❖ versatile peroxidase (VP);
❖ dye-decolourising peroxidase (DyP). As a special group, extract peroxidases and explain why,
for both economic, engineering and biotechnological reasons, it is necessary to do production by
gene mutation, using E.coli. The part od Introduction that describes dye-decolorizing peroxidases
is very well; keep it.

The main objective must to be clarified and accentuated more clearly, so that it is clear to the
reader at the very beginning why the introduction of a new loop around the heme pochet have is
better.

Answer: We have removed the first paragraph and started the Introduction as the reviewer suggested “Biocatalysis is both a green (…)”. Here we have highlighted the importance of lignin oxidative transformation (lines 39 to 47) and added the classification of heme peroxidases involved in lignin conversion (lines 48 to 50). The main objective of the work was emphasized in the last paragraph of the Introduction, as follow: “The properties of evolved variants based in their biochemical, kinetic and structural analysis are discussed. In DyPs, oxidation of reduced substrates occurs in heme cavities, and in tyrosine and tryptophan surface-exposed residues, similarly to LiP and VP enzymes, and then to transfer electrons to the heme using long-range electron transfer (LRET) pathways [16, 27, 28]. However, details of substrate binding and of molecular determinants of substrate specificity in DyPs remain open questions. This work helps understanding the role of conserved loops around the heme pocket in substrate binding and catalysis, and the interplay of catalytic and stability mechanisms of DyPs with implications in their industrial application and in the future design of enzymes.” 

#5 – The methods are generally appropriate and well written, but some additional informations are necessary so that the procedures can be replicated by other researchers. Therefore, it is necessary for the authors to state the names of companies/corporations, but also their headquarters (names of cities and names of countries) from which the devices were purchased. Throughout the whole section Materials and Methods of the manuscript this is necessary.

Answer: The information on details of devices was added to the Materials and Methods section. 

#6 – Why authors did not describe the methods which are used for the ion-exchange (SPSepharose) and size-exclusion (Superdex-75) chromatography, that were used during the purification steps of peroxidase (BsDyP) after its production ande separation from fermentation medium? It is mandatory to write all methods. Please, write this part in the manuscript like special section, like as Purification od prodduced BsDyP, which will contain the description of separation steps from raw enzyme exctrat to purified BsDyP. Ion-exchange (SP-Sepharose) and sizeexclusion (Superdex-75) chromatography steps have been seen in the Supplementary Figures of SDS-PAGE electropherogram.

Answer: Following the reviewer request, the description of purification methods was added to Materials and Methods section, (page 14, lines 520 to 526).

#7 – How authors calculate enzyme activity? Please write the method by which the enzyme activity is determined. This implies defining the unit of enzymatic activity (IU/ml or IU/mg), ie.  accurate statement of what is mean 1 IU of peroxidase enzyme, determined under the tested reaction conditions.

Answer: In pg 14, lines 502 to 707: “To test the enzymatic activity of variant libraries, 20 µL of cell crude extracts were transferred to 96-well plates containing 180 µL of reaction mixture (100 mM sodium acetate, pH 3.8, 1 mM DMP and 0.2 mM H2O2). The progress of reactions was monitored at 468 nm (ε498 = 49,600 M−1 cm−1) using a Synergy2
microplate reader (BioTek, Winooski, Vermont, USA). One unit (U) of enzymatic activity was defined as the amount of enzyme required to reduce 1 µmol of the substrate per minute.”

#8 – Is the Michaelis-Menten kinetic model sufficiently adequate to fit the experimental
results of the kinetic study? It is a two-substrate enzymatic reaction, ie. a reaction in which two
substrates are required to determine the adequate kinetic parameters, as well as the BsDyP activity:
1) hydrogen peroxide and 2) ABTS/DMT in your manuscrit. Please pay attention to the above, check the results of the kinetic model.

One note, two-substrate enzymatic reactions are modeled by so-called ping-pong bi-bi kinetic
models, which take into account the concentrations of both substrates and are derived from the
Michaelis-Menten equation. Keep this in mind when checking.

Answer: The Michaelis-Menten kinetic model fits the experimental result with an R2 value of circa 0.98, for this reason we believe it is adequate to represent the enzymatic reactions. As pointed-out by the reviewer, DyPs catalyze a two-substrate enzymatic reaction following the so-called ping-pong bi-bi kinetic model. The kinetic parameters were estimated just considering the variation of one of the substrates while maintaining the second substrate at a saturating concentration. Therefore, the formalism is to mention these as “apparent steadystate kinetic parameters”, this was already in the Tables ledend and in Material and Methods, we have additionally reported kcat as kcat app and km as

Km app thorough the manuscript. 

#9 – Authors defined the BsDyP specific activity on the Figure S5 as U (unit), but in the Table S3, the same activity is defined as mU (mili unit). What is the reason for this difference? Is it a mistake? If there is no error, be sure to clearly indicate in the sections of methods where necessary how the results will be expressed/defined. This creates confusion and gives the impression of misinterpreting the results.

Answer: In table S3 the specific activities were measured in crude extracts, and were too low to be represented in U/mg and were therefore represented in mU/mg. In figure S5 the specific activity was measured with purified enzyme preparations, the obtained values are significantly higher and were represented in U/mg. 

#10 – Conclusion is acceptable in this form. Readability of paper satisfactory. The technical preparation of the manuscript according to the section Results and discussion is satisfactory. The results were interpreted in a critical way, and compared with valid literature data.

Answer: We thank the reviewer for the positive comments.

#11 – Literature study: sufficient and good amount of information is included.

Answer: We thank the reviewer for the comments and suggestions, which were important contributions to improve the quality of the manuscript. 

Reviewer 2 Report

Rodrigues and Borges et al have used E.coli-based directed evolution to identify evolved Bacillus subtilis dye-decolorizing peroxidase (BsDyP) variants for improved oxidation of a model lignin-derived phenolic. BsDyP exhibits activity on various lignin-derived phenolic compounds and high redox potential synthetic dyes, and the native BsDyP possesses very good thermal stability and is highly pH-stable. However, the activity of BsDyP for phenolics is less satisfactory for potential industrial applications. The authors now obtain a BsDyP variant with seven fold higher catalytic rates as compared to the wild type enzyme after three rounds of evolution. They also notice the enhanced protein yields by a factor of two for some of the variants. Crystal structures of wild type BsDyP and the 5G5 mutant highlight that the loops and a small helix around the heme pocket are critical regulators of the catalytic turnover rate, Michaelis constant of BsDyP towards substrates, as well as the chemical and thermal stability of the enzyme.

Strengths:

The findings are interesting and shed new light on function regulation of DyP by flexible loops around (and also shaping) the substrate tunnel and enzyme active site. In particular, they provide evidence that the reported activity enhancing missense mutations are all localized closely to the heme pocket, which may provide a general scheme for protein engineering of DyPs from other species.

Weaknesses:

This study has no major weaknesses. Thus, some minor suggestions are listed below.

  1. It would be beneficial to the readers if the authors would expand the introduction section by comparing the substrate specificity of DyPs from different classes. Figure 5 clearly shows the molecular tunnels and cavities are different among the 3 Classes of DyPs. And Figure 4 illustrates structural rearrangements that resulted from the K317E substitution. Do structural rearrangements alter the substrate specificity of BsDyP in addition to the altered Km?
  2. Section 3.7 (Line 500 - 508). The total protein concentration in cell crude extracts was measured using Bradford assay. Did the authors consider the different expression level for each variant when calculating specific activity (Table S1)?
  3. Line 513. Please describe the purification method briefly, instead of just referring to a paper.
  4. Figure 1. Please indicate which variant is the parent for 5G5.
  5. Figure 2. To be consistent, It is better to label “A330” vs “A330V” , since K317 is not visible in 5G5 and the authors only label “K317”. Same for S325P.
  6. Table 2. Please draw a vertical line to divide the data for H2O2 from the data for ABTS.
  7. Table 3. Please indicate the concentration of H2O2 used for 54D6.
  8. Table 5. Please indicate number of replicates.
  9. Figure S2. Please include the rationals to choose 2 and 1.5 as thresholds. The authors may also include data for wild type enzyme in S2Ab.
  10. Figure S4. The figure is a little bit busy. Is it possible to use separate panels for every profile?
  11. Figure S9. Please describe the inset figure.

Author Response

Weaknesses:

This study has no major weaknesses. Thus, some minor suggestions are listed below.

1.It would be beneficial to the readers if the authors would expand the introduction section by comparing the substrate specificity of DyPs from different classes. Figure 5 clearly shows the molecular tunnels and cavities are different among the 3 Classes of DyPs. And Figure 4 illustrates structural rearrangements that resulted from the K317E substitution. Do structural rearrangements alter the substrate specificity of BsDyP in addition to the altered Km?

Answer: At the moment and in our opinion there is not a sufficient body of information that clearly point to a class-dependent DyPs substrate specificity. There are however clear indications that members from class V show higher catalytic efficiencies (kcat/Km) and this is in the ms, pg 2, line 62 (“Members from this class are in general the most efficient catalysts [16-18].”). Our data show that the structural rearrangements increased the Km for  both ABTS and DMP, putatively implying that K317 substitution affected their binding site, and indicating that these relatively small substrates are oxidized close to the heme cavity. We did not test further substrates.

2. Section 3.7 (Line 500 - 508). The total protein concentration in cell crude extracts was measured using Bradford assay. Did the authors consider the different expression level for each variant when calculating specific activity (Table S1)?

Answer: The expression levels as assessed by SDS-PAGE were comparable in crude extracts of cells producing the different variants therefore we did not took into consideration different ratios of protein production.

3. Line 513. Please describe the purification method briefly, instead of just referring to a paper.

Answer: Following the reviewer request, the description of purification methods was added to Materials and Methods section, (page 14, lines 520 to 526). 

4. Figure 1. Please indicate which variant is the parent for 5G5.

Answer: Variant 5G5 results from the DNA Shuffling which recombined the genes from variants: 3G5 (from the first generation) and 50F7, 51E9, 54D6 and 56C8 (from the second generation). However, for the screenings purpose we have used the activity and stability of variant 50F7 as reference.  

5. Figure 2. To be consistent, It is better to label “A330” vs “A330V” , since K317 is not visible in 5G5 and the authors only label “K317”. Same for S325P.

Answer: We agree with the reviewer and a new figure was prepared with the modifications suggested. 

6. Table 2. Please draw a vertical line to divide the data for H2O2 from the data for ABTS.

Answer: Following the reviewer request, the table was modified accordingly. 

7. Table 3. Please indicate the concentration of H2O2 used for 54D6.

Answer: Following the reviewer request, the concentration of H2O2 (0.6 mM) used for 54D6 kinetics was added (Table 3). 

8. Table 5. Please indicate number of replicates.

Answer: The number of replicates were three and this information was added in table 5 (page 11, line 381).

9. Figure S2. Please include the rationals to choose 2 and 1.5 as thresholds. The authors may also include data for wild type enzyme in S2Ab.

Answer: The thresholds were defined considering the coefficient of variance (CV) calculated for the parent of each generation of variants. The CV calculated with wild-type enzyme was circa 25%, while for 1F9 and 50F7 (parents of the second and third generations, respectively) was circa 20%. This led the authors to consider a wider interval of confidence when choosing the variants with increased activity in the first generation and decreasing that interval in the generations that followed. 

10. Figure S4. The figure is a little bit busy. Is it possible to use separate panels for every profile?

Answer: We agree with the reviewer, but we had opted for showing the pH profiles of wild-type and variants together to emphasize their similarity. 

11. Figure S9. Please describe the inset figure.

Answer: Following the reviewer request, the description of the inset figures was added, to the legend, as follows: “The inset figures show that the activity decay of BsDyP wild-type and variants can be fitted to a single first-order process, as the logarithm of activity displays an inverse linear relationship with time.” 

We wish to thank the reviewer for the comments and suggestions, which were important contributions to improve the quality of the manuscript.